# The Impact of Undergraduates’ Social Isolation on Smartphone Addiction: The Roles of Academic Anxiety and Social Media Use

**DOI:** 10.3390/ijerph192315903

**Published:** 2022-11-29

**Authors:** Youlai Zeng, Jiahui Zhang, Jiaxin Wei, Shunyu Li

**Affiliations:** 1School of Education, Liaoning Normal University, 850 Huanghe Road, Dalian 116029, China; 2Center for Teacher Education Research in Xinjiang, Xinjiang Normal University, 100 Guanjing Road, Urumqi 830017, China

**Keywords:** social isolation, social media use, academic anxiety, smartphone addiction

## Abstract

The COVID-19 pandemic has an adverse effect on the physical health of societies and individuals. One important concern is the effect of social isolation on the mental health of undergraduates, such as academic anxiety, smartphone addiction and other social psychological problems. The purpose of this study was to investigate associations among undergraduates’ social isolation in this special context, social media use for obtaining information about the COVID-19 pandemic (i.e., communicative and non-communicative), academic anxiety, and smartphone addiction. A cross-sectional survey was conducted from May to June in 2022 and a total of 388 undergraduates were included. The results showed significant positive associations between social isolation and smartphone addiction and academic anxiety. Furthermore, academic anxiety played a mediating role in the effect of social isolation on smartphone addiction, which was moderated by non-communicative social media use. Some theoretical and practical implications as well as research limitations are discussed.

## 1. Introduction

The COVID-19 pandemic has caused undergraduates to experience varying degrees of anxiety [1], and has led to a growing sense of social isolation [2]. In this context, smartphones have become an important medium for undergraduates to obtain information about the pandemic and have provided many functions such as games, entertainment and social networking. While smartphones bring convenience, they also bring some social health problems, such as smartphone addiction [3]. There are many studies on the antecedent of problematic smartphone use behavior among undergraduates and its relationship with anxiety [4,5]. However, the number of studies exploring the effect of social media use and academic anxiety on the influencing process between social isolation and smartphone addiction among undergraduates is small. The present study attempts to explore the influencing mechanism of social isolation on smartphone addiction. On the one hand, this research extends the literature by exploring the influence mechanism of smartphone addiction from the perspective of social isolation as a pre-factor. On the other hand, in the context of the COVID-19 pandemic, the test of the effects of social media use and academic anxiety on the relationship between social isolation and smartphone addiction can further expand the possible factors influencing smartphone addiction, and also draw attention to the physical and mental health of undergraduates.

### 1.1. Undergraduates’ Social Isolation and Smartphone Addiction

#### 1.1.1. Social Isolation

Social isolation was first proposed by Berkman and Syme in 1979 [6], they argued that the main characteristic of social isolation is the lack of social ties and resources for individuals, and marriage, contact with close friends and relatives, church membership, and informal and formal group associations of four social resources were used to measure an individual’s level of social isolation. Since the concept of social isolation was proposed, it had received widespread attention from scholars, but its connotations have been defined inconsistently [7]. In summary, the connotations of social isolation can be divided into two types. One is the objective level of social isolation. Scholars initially interpreted social isolation according to its literal meaning, mainly referring to the state of isolation between people [8]. Kim believed that physical distancing between people was also an objective social isolation, and measured social isolation by the degree of physical distance retention [9]. The other type is social isolation at the subjective level, which refers to some psychological states in which an individual’s interaction with others lacks social belonging, true engagement, and fulfilling relationships [10]. Some scholars used social network scales to measure subjective social isolation [11], that is, if a person scores low, they are considered to have social isolation. Another scholar measured it with the loneliness index, which can more directly reflect the degree of social isolation of individual subjective perception [12]. Since the outbreak of the COVID-19 pandemic, many countries have introduced movement restrictions to achieve good prevention and control results [13]. However, some problems must be considered, for example, social isolation. A survey of 14,302 adults from 101 countries showed a 13% prevalence of social isolation in 2021 [14], another study of 32,359 Spaniards yielded 26% in 2022 [15]. The prevalence of social isolation is increasing as the COVID-19 pandemic continues. During the COVID-19 pandemic, the biggest challenge for undergraduates is reduced interaction with classmates and teachers, with surveys showing that 70% of undergraduates’ studies or work is negatively affected [16]. Therefore, the respondents of this study mainly focus on the college student group, and define social isolation as two dimensions: loneliness at the subjective level and social distancing at the objective level.

#### 1.1.2. Smartphone Addiction

Addiction is defined as a problematic behavior that occurs repeatedly and continues despite knowing it has negative consequences [17]. Griffiths has proposed the symptoms of addiction, including relapse, conflict, withdrawal symptoms, mood modification, salience, and tolerance [18], which are widely recognized and used in academia. There is no consensus in the academic community as to whether smartphone addiction can be considered an addiction. Some scholars contend that smartphone addiction is not an addiction, because there is no evidence that smartphone addiction has neurobiological and psychological mechanisms similar to other chemical addictions or behavioral addictions (internet, gambling addiction) [19], nor is it one of the officially defined addiction disorders [20]. Some other scholars believe that smartphone addiction is an addiction, for example, Bianchi et al. refer to the theory of addiction to study smartphone addiction, and believe that it is a non-chemical human–computer interaction behavior addiction [21]. There are also scholars who learn from the theory of internet addiction to define the connotation of smartphone addiction [22,23], and insist that it has cognitive salience, loss of control, mood modification, tolerance, withdrawal, conflict, and relapse and other behavioral addiction manifestations. Smartphone addiction as an addiction has been widely recognized by scholars, and with the popularity of smartphones, more scholars have drawn from internet addiction theories to study smartphone addiction [24,25]. At the same time, these scholars have also developed a specialized smartphone addiction assessment tool. The most widespread of these assessment tools is the smartphone addiction scale adapted by Kwon et al. on the basis of the internet addiction scale [26], which contains 33 questions in six dimensions of daily life disturbance, positive anticipation, withdrawal, cyberspace-oriented relationship, overuse, and tolerance, and the internal consistency coefficient is 0.967. Subsequently, in order to more easily assess smartphone addiction, Kwon et al. further streamlined the scale, forming a condensed version of the smartphone addiction scale with 10 questions [27] which has been widely used in different countries [28,29]. Luk and other scholars have developed a Chinese version of the smartphone addiction scale, which has an internal consistency coefficient of 0.84 [30], and it is well adapted in the context of Chinese culture. Smartphone addiction is becoming increasingly prominent as a public health problem, as a recent survey of 7531 adults in 14 countries showed that the prevalence of smartphone addiction was 28.1% [31]. Some scholars have found that the proportion of smartphone addiction among undergraduates during the COVID-19 pandemic is 37–51% [32,33]. Compared with other groups, undergraduates tend to have smartphone addiction [34]. Therefore, it is necessary to study the factors and mechanisms of smartphone addiction of undergraduates.

In recent years, with the outbreak of the COVID-19 pandemic, some scholars have begun to explore the formation mechanism of smartphone addiction [32,35], especially the impact of new individual and environmental changes due to the COVID-19 pandemic, for example, the impact of social isolation on smartphone addiction. Some studies have shown that the social isolation status of individuals cannot predict the level of smartphone addiction, and there is no correlation between social isolation and smartphone addiction at the objective level [36]. However, some other studies have shown that social isolation can significantly predict the level of smartphone addiction, and that loneliness at the subjective level of social isolation in undergraduates is a direct predictor of smartphone addiction [37]. This partly confirms Durkheim’s theoretical view that large-scale social crisis can take a toll on an individual’s health and well-being by weakening social integration [38]. The COVID-19 pandemic outbreak may increase social isolation to influence individual health and well-being. Individuals with low happiness and low health tend to develop smartphone addiction [39]. However, there are few empirical studies on the relationship between social isolation and smartphone addiction in the context of COVID-19. Consequently, this study speculates that the COVID-19 pandemic may increase social isolation, and that undergraduates tend to keep in touch with others by smartphones to distract themselves, which will make the students more likely to become addicted to smartphones. As such, the current research tries to use empirical research methods to examine the relationship between social isolation and smartphone addiction.

Based on this, this study proposes the following hypothesis known as
**H1:** *Social isolation among undergraduates has a positive effect on smartphone addiction.*

### 1.2. The Relationship between Undergraduates’ Social Isolation and Smartphone Addiction: The Role of Academic Anxiety and Social Media Use

#### 1.2.1. Academic Anxiety

Academic anxiety refers to feelings of worry, nervousness, and uneasiness associated with achievement in school environment [40,41]. In the light of the theory of achievement emotion [42,43], academic anxiety includes learning anxiety, classroom anxiety and test anxiety, which are also the indicators applied in this study to define the academic anxiety of undergraduates. Academic anxiety among undergraduates is a very common phenomenon, and some studies in particular have shown that the COVID-19 pandemic has exacerbated it [44]. Therefore, how to reduce and control the level of academic anxiety of undergraduates and explore its causes are the areas of focus in the research field. Recently, scholars have found that social isolation caused by the COVID-19 pandemic is an important factor in anxiety [12,45], but few scholars have explored whether social isolation also has a direct impact on academic anxiety.

Numerous studies have confirmed that anxiety in undergraduates is significantly connected with smartphone addiction [46,47], which supports the problem-behavior theory. According to the theory, negative emotions aggravate an individual’s problematic behavior [48], when individuals have negative emotions such as anxiety, it will aggravate the emergence of problematic behavior such as smartphone addiction. Individuals with anxiety may overuse their smartphones to alleviate emotional distress [49], resulting in more problematic smartphone use behavior [50]. The COVID-19 pandemic has caused anxiety to persist [51,52], which may enhance the level of smartphone addiction in undergraduates. As research goes deeper, scholars begin to focus on how academic anxiety levels affect smartphone addiction [53]. Therefore, this study infers that in the context of the COVID-19 pandemic, social isolation may lead to higher levels of academic anxiety, which is bond up with smartphone addiction, and there may be a certain influence on the process of social isolation affecting smartphone addiction among undergraduates.

Based on this, this study proposes the following hypothesis known as:
**H2:** *Academic anxiety in undergraduates has a mediating effect between social isolation and smartphone addiction.*

#### 1.2.2. Social Media Use

Social media use refers to a set of internet applications based on Web 2.0 ideas and technologies that supports the generation and exchange of user content, allowing people to write, share, evaluate, discuss, and communicate with each other [54], via typical applications such as Facebook, Twitter, WeChat, and Weibo, etc. Chan divides smartphone use into two types, including communicative and non-communicative [55], and Stevic applies these two types to social media use to understand the individual’s access to information about the COVID-19 pandemic, and communicative social media use emphasizes the acquisition of relevant information through communication with others, while non-communicative social media use refers to the acquisition of information by personally browsing news, websites, etc. [56]. With the outbreak of the COVID-19 pandemic, scholars have begun to pay attention to the relationship between social isolation and social media use, but no consensus has been reached. Some scholars’ findings suggest that frequent use of social media and social isolation are significantly correlated [57], but some other scholars have found that social media use time is not related to social isolation [58]. Relatively speaking, the research conclusions on the relationship between social media use and anxiety are more consistent, that is, undergraduates with problematic social media use showed significantly higher anxiety levels [59]. In addition, scholars have also explored the relationship between social isolation, social media use, and anxiety. Boursier, for example, found that social media overuse mediates between social isolation and anxiety [60]. There are also studies that confirm that the way individuals use social media in the context of the COVID-19 pandemic is significantly correlated with individual psychological distress [61]. However, no scholars have studied the influence of communicative and non-communicative social media use between social isolation and academic anxiety. Therefore, the present study further speculates that the social media use (communicative/non-communicative) when undergraduates obtain news of the COVID-19 pandemic and social isolation may interact with academic anxiety.

In the context of the COVID-19 pandemic, undergraduates, whether they use communicative social media or non-communicative social media to obtain messages about the pandemic, will expose themselves to the stressors of the pandemic, leading to anxiety [62]. This relates to the theory of the stress process model that holds that stressors, stress, and distress influence one another [63]. The stress caused by environment and experiences is difficult to adjust or change and can have a detrimental effect on an individual’s mood, cognition, behavior, physical functioning, and well-being [64]. It is worth further exploring the process of social isolation affecting smartphone addiction levels through academic anxiety, and different social media use may have different effects on the process. As some scholars have found, non-communicative social media use in the context of COVID-19 has a negative impact on individual physical and mental health [65], while communicative social media use has a positive impact [56]. Only proper use of social media can alleviate individual anxiety and mitigate the negative effects of social isolation [66]. Therefore, this study speculates that there are certain differences in social isolation, academic anxiety and smartphone addiction among undergraduates who adopt different social media use when obtaining information about COVID-19 pandemic, which means that the impacts of social isolation on smartphone addiction through academic anxiety are different.

Based on this, we have formulated hypothesis:
**H3:** *Social media use (communicative and non-communicative) when undergraduates obtain news of COVID-19 has a moderating effect on the relationship between social isolation and academic anxiety, and moderate the mediating effect of academic anxiety between social isolation and smartphone addiction.*

## 2. Research Methods

### 2.1. Participants

In this study, a cross-sectional survey was conducted from May to June in 2022 from a normal university in Liaoning Province for statistical analysis. In the spring semester of 2022, 14 teacher-training majors such as Chinese international education, English, and mathematics were selected as the sampling population, including 6380 undergraduates, which covered all second-level colleges, and the purpose of hierarchical sampling was basically achieved. The testing process was carried out in the form of a combination of the participation of the researcher themselves and the entrusted agent (public course teachers) through the distribution of an online questionnaire to the class teaching and learning WeChat group, and all the participants were told that the questionnaire survey is conducted in an anonymous form, and the data collected are used completely for scientific research with strict confidentiality.

A total of 345 questionnaires were obtained, accounting for 5.41% of the total population size, 7 invalid questionnaires with regular responses were excluded, finally 338 valid questionnaires were obtained (73 males and 265 females), and the questionnaire recovery efficiency was 97.97%. The participants were mainly sophomore and juniors, including 26 in the first year, 90 in the sophomore, 163 in the third year, and 59 in the senior year. The mean age of participants was 20.56 years, of which 59 were under 19 years old, 214 were 20–21 years old, and 65 were over 22 years old.

*t*-test results showed that in terms of gender, social isolation (t = 1.99, df = 336, *p* = 0.05), communicative social media use (t = −0.65, df = 336, *p* = 0.52), non-communicative social media use (t = −1.64, df = 336, *p* = 0.10), academic anxiety (T = 0.2, df = 336, *p* = 0.98), smartphone addiction (t = 1.23, df = 336, *p* = 0.22), there is no significant difference. The results of one-way ANOVA showed that in different academic stages, the results of social isolation (F = 1.22, *p* = 0.30), communicative social media use (F = 2.59, *p* = 0.05), academic anxiety (F = 1.12, *p* = 0.34), smartphone addiction (F = 0.17, *p* = 0.92) and homogeneity of variance test were all homogeneous. The results of LSD were not significant (*p* > 0.05), indicating that there was no difference between these variables at different academic levels. Non-communicative social media use (F = 3.49, *p* = 0.02) showed uneven homogeneity of variance test results, and Tamhane’s multiple comparison results showed differences in different academic levels.

### 2.2. Measures

#### 2.2.1. Social Isolation

Our main predictor is social isolation including social distancing (2 items) and loneliness (6 items). Concerning the COVID-19 pandemic, two questions inquire about social distancing referring to Kim’s study: “To what extent do the following statements describe your behavior for the past week? A. I did not attend social gatherings; and B. I kept a distance at least one meter to other people.” The original answers, coded on a scale ranging from 1 (“Does not apply at all”) to 6 (“Applies very much”) [9]. Loneliness was measured by a 6-item scale (RULS-6), such as “How often do you feel that you lack companionship?”, “How often do you feel lonely?”. The participants make a judgment on a 6-point Likert scale. Cronbach’s alpha of the RULS-6 was 0.83 [67]. A Cronbach’s alpha of 0.94 was obtained in the present study. The sum of the eight questions’ score represents the level of social isolation, and a high score implies a high perceived level of social isolation while a low score indicates a low level.

#### 2.2.2. Smartphone Addiction

The outcome variable is smartphone addiction, which is measured by the Chinese version of Smartphone Addiction Scale-Short Version (SAS-SV), a single factor structural scale including 10 items, and the Cronbach’s alpha of the scale was 0.84 [30]. The participants make a judgment on a 6-point Likert scale (1 = strongly disagree; 6 = strongly agree) on the 10 items, such as “Missing planned work due to smartphone use” and “The people around me tell me that I use my smartphone too much”. A Cronbach’s alpha of 0.92 was obtained in the present study.

#### 2.2.3. Social Media Use

The moderating variables are types of social media use for accessing information about the COVID-19 pandemic, which was measured by Stevic’s [56] smartphone use scale, a two factors structural scale including 6 items. For example, the participants were asked (on a 6-point Likert scale; 1 = never; 6 = very often) how often did they use their smartphone last week to talk to family members, friends, or acquaintances about the COVID-19 pandemic. In the present study, the internal consistency reliability coefficient of communicative social media use scale is 0.84 and the non-communicative social media use scale’s is 0.86.

#### 2.2.4. Academic Anxiety

The Emotional Achievement Questionnaire (AEQ) is a multi-dimensional self-reporting tool designed to assess a college student’s achievement sentiment and examine the emotions that students experience in the context of academic achievement [68], which can be used to assess and measure classroom-related, learning-related, and exam-related emotions, including enjoyment, hope, pride, anger, anxiety, shame, despair, and boredom. According to the cognitive, emotional, mental, physiological dimensions and academic activity time, the corresponding questions were extracted from the three scales of classroom anxiety, learning anxiety and test anxiety, respectively, for a total of 9 questions. In short, the academic anxiety scale consists of 3 dimensions and 9 questions. Judgment is made on 6 degrees, 1 = completely inconsistent, 6 = very consistent. For example, “Thinking about class makes me feel uneasy”. “I get tense in class”. “When I look at the books I still have to read, I get anxious”. “I feel panicky when writing the exam”. A Cronbach’s alpha of 0.93 was obtained in the present study. The sum of the 9 questions’ score represents the level of academic anxiety, and a high score implies a high perceived level of academic anxiety while a low score indicates a low level.

## 3. Results

### 3.1. Descriptive Statistics and Correlation Analysis

Table 1 reports the mean, standard deviation, and the correlation coefficient for each variable. From the analysis results, social isolation is significantly positively correlated with smartphone addiction (r = 0.454, *p* < 0.01) and academic anxiety (r = 0.502, *p* < 0.01). Communicative social media use was significantly positively correlated with non-communicative social media use (r = 0.709, *p* < 0.01), smartphone addiction (r = 0.394, *p* < 0.01), academic anxiety (r = 0.257, *p* < 0.01), and social isolation (r = 0.176, *p* < 0.01). Therefore, this result provides preconditions for subsequent hierarchical regression analysis.

### 3.2. Common Method Biases Testing

Table 2 reports the Harman one-way test statistics, before the rotation of the factor analysis extracted a total of six factors with eigenvalues greater than 1, the cumulative variance contribution rate reached 71.70%, of which the first (the factor with the largest eigenvalue) factor variation was 36.91%, less than 40%; the maximum eigenvalues of the six factors after rotation were 5.66, and the cumulative variance contribution is 17.16%, which was less than 20%. Before and after the rotation, there was no phenomenon of a factor explaining most of the variations, which can indicate that the common method bias test results are good, and the common method bias had no serious effect, and did not affect the use of relevant data and the test of relevant hypotheses in this study.

### 3.3. Hypothesis Testing

In the first place, hierarchical regression analysis was applied to verify the direct impact of social isolation on smartphone addiction among undergraduates. The results indicated that after controlling demographic variables such as gender, age, grade and major, social isolation had a significant positive effect on smartphone addiction (b = 0.55, *p* < 0.01).

Then, the SPSS macro-program PROCESS (Guilford Publications, Inc, New York, NY, USA) developed by Hayes was used to analyze the mediating effect of academic anxiety between social isolation and smartphone addiction (see Table 3). The results of statistical analysis showed that after controlling the demographic variables, social isolation had a significant positive effect on academic anxiety (M1, b = 0.55, *p* < 0.01), and after placing social isolation and academic anxiety in the regression model at the same time, the results showed that the effect of academic anxiety on smartphone addiction was significant (M2, b = 0.61, *p* < 0.01). The non-parametric percentile Bootstrap method of deviation correction was further used to test the mediating effect of academic anxiety. The results showed that in the path of social isolation affecting smartphone addiction, the mediating effect of academic anxiety was significant, the indirect effect value was 0.33, accounting for 61% of the total effect, Boot 95% confidence interval was between 0.25 and 0.44, and the confidence interval did not contain 0, indicating that the mediating effect of academic anxiety was significant, and Hypothesis 2 was verified.

Finally, this study continues to use macro-program PROCESS to analyze the moderating effect of social media use (communicative/non-communicative) on the mediating effect of academic anxiety. The results of the analysis showed (see Table 4) that the effect of social isolation and non-communicative social media use on academic anxiety in the regression model reached a significant level (M3, b = 0.03, *p* < 0.01), that is, the non-communicative social media use had a significant effect on the association between social isolation and academic anxiety.

To more intuitively characterize the moderating effects of the relationship between social isolation and academic anxiety, the researchers plotted a simple slope of the moderating effects (see Figure 1). According to this, when undergraduates demonstrate more non-communicative social media use, the positive relationship between social isolation and academic anxiety is stronger; when undergraduates demonstrate less non-communicative social media use, the positive association between social isolation and academic anxiety is weaker.

To further examine the moderating role of non-communicative social media use on the mediating effect of academic anxiety (see Table 5). The results showed that when undergraduates had less non-communicative social media use, the indirect effect of social isolation on smartphone addiction through academic anxiety was weak (ρ = 0.20, Boot 95% CI does not contain 0); when undergraduates had more non-communicative social media use, the indirect effect of social isolation on smartphone addiction through academic anxiety was stronger (ρ = 0.33, Boot 95% CI does not contain 0). On the basis of the moderating mediating effect test proposed by Hayes (2015), the test parameter index is 0.02, and the Boot 95% confidence interval is between 0.001 and 0.029, which does not contain 0. In conclusion, the mediating effect of academic anxiety on the relationship between social isolation and smartphone addiction is moderated by non-communicative social media use, that is, there is the moderating mediating effect.

The results of the analysis showed (see Table 6) that the effect of social isolation and social media use in the regression model on academic anxiety did not reach a significant level (M5, b = 0.003, *p* > 0.05), that is, the moderating effect of social media use on the relationship between social isolation and academic anxiety was not significant. In line with the moderating mediating role test proposed by Hayes (2015), the test parameter Index is 0.002, and the Boot 95% confidence interval is between −0.02 and 0.02, which contains 0. In conclusion, the mediating role of academic anxiety on the association between social isolation and smartphone addiction is not moderated by social media use, that is, there is no moderating mediating effect.

So far, hypothesis H3 has been partially validated. Non-communicative social media use has a moderating role on the association between social isolation and academic anxiety, and moderates the mediating effect of academic anxiety between social isolation and smartphone addiction in undergraduates, while communicative social media use does not have such a moderating effect.

## 4. Discussion

### 4.1. Social Isolation and Smartphone Addiction

Hypothesis H1 has been confirmed in present study. Social isolation among undergraduates has a positive effect on smartphone addiction. Social isolation due to COVID-19 plays a significant role in the emergence of problematic internet use, social media addiction and problematic video gaming [2,69]. The conclusions of this study showed that social isolation among undergraduates had a positive effect on smartphone addiction. This result is consistent with the findings of previous studies, which confirm the negative effects of social isolation in the context of COVID-19. In the context of the COVID-19 pandemic, the stronger the sense of social isolation and loneliness, the more likely undergraduates are to become addicted to mobile phones. This conclusion and the findings of Gregorio have been unanimous [70], due to social isolation measures, Italian children and adolescents showed more frequent smartphone use during the COVID-19 pandemic compared to the pre-pandemic period. This finding shows no difference with Al-Kandari’s regression analysis, the higher the level of social isolation, the greater the chance of smartphone addiction [3]. Furthermore, the operational connotations of social isolation in this study include loneliness at the subjective level, and many studies have confirmed that loneliness caused by COVID-19 restrictions is an important factor of smartphone addiction [71,72]. The conclusions of this study are also indirectly in line with them, which further validates the problem-behavior theory.

The conclusions of this study suggest that, on the one hand, we should take measures to alleviate the social isolation of undergraduates owing to the COVID-19 pandemic, and on the other hand, we also need to find a method of breaking away from the problematic smartphone use behavior itself to reduce the risk of smartphone addiction. Protection Motivation Theory (PMT) has demonstrated that fear influences individuals to seriously engage in protective behaviors to minimize or avoid threats [73]. Therefore, low academic achievement caused by smartphone addiction [74,75] can be regarded as a threat or fear as well as adverse physiological reactions such as numbness in the hands and feet, palpitations, and gastrointestinal dysfunction. According to this, in the educational psychological intervention, attention should be paid to clarify the negative effects of problematic mobile phone behavior, such as smartphone addiction, in undergraduates, thereby reducing the incidence of smartphone addiction.

### 4.2. The Mediating Role of Academic Anxiety

Hypothesis H2 has been confirmed in present study. Academic anxiety in undergraduates has a mediating effect between social isolation and smartphone addiction. The effect of social isolation on smartphone addiction among undergraduates is partly achieved through academic anxiety. This conclusion is consistent with cognitive theories of stress and coping [76], if people perceive a dangerous situation and do not have enough capacity to deal with the threat, they develop a stress response that includes overt anxiety and attempts to escape from the situation.

To begin with, this conclusion is consistent with existing research that academic anxiety will be caused by uncertainty and other negative factors in social isolation [77,78], undergraduates with stronger social isolation will show higher levels of academic anxiety. This finding is also consistent with Durkheim’s sociological theory that large-scale social crisis can harm an individual’s health and well-being by destroying social integration. This finding is also consistent with Bradburn’s theoretical perspective that negative emotions are a symptom of an individual’s negative mental state (such as loneliness) and other emotional disorders [79]. Under the COVID-19 pandemic, experiences of social isolation such as loneliness and distance among undergraduates tend to lead to negative emotions such as academic anxiety.

This finding further expands the perspective of the relationship between anxiety and smartphone addiction. Serious emotional symptoms (anxiety) are risk factors for undergraduates’ smartphone addiction behavior [80], social anxiety is significantly related to problematic mobile phone use behavior [50]. This study confirmed the positive effect of academic anxiety on smartphone addiction and academic anxiety is a risk factor for smartphone addiction. This study suggests that there is a need to further explore the mediating variables that interact between academic anxiety and smartphone addiction behavior. Just as fear of missing out mediates between individual anxiety and smartphone addiction [81], similarly, other tests of the role of subjective variables in undergraduates will have implications for interventions on smartphone use and other problems.

In the end, the confirmation of the mediating role of academic anxiety further clarifies the mechanism of the relationship between social isolation and smartphone addiction of undergraduates. In line with the view, due to the COVID-19 pandemic, people are at risk of falling into a negative mental health spiral that can also affect their performance over time [82]. According to this research result, academic anxiety, as a mediating variable in the impact of social isolation on smartphone addiction, should prompt educators to pay full attention to the academic anxiety of undergraduates in the context of the pandemic, take effective measures to reduce academic anxiety, and thus minimize the risk of social isolation leading to smartphone addiction.

### 4.3. The Moderating Role of Social Media Use

Hypothesis H3 has been partially validated. Non-communicative social media use when undergraduates obtain news of COVID-19 has a moderating effect on the relationship between social isolation and academic anxiety, and moderates the mediating effect of academic anxiety between social isolation and smartphone addiction, but communicative social media use does not have this effect.

On the one hand, non-communicative social media use regulates the relationship between social isolation and academic anxiety in a positive way. That is, the more undergraduates obtain information about the pandemic through non-communicative social media use, the greater the likelihood of academic anxiety caused by social isolation. This conclusion and existing findings unanimously demonstrate that anxious participants tend to use social media more often to actively search for a way of adapting to the current situation, and to a lesser extent as a way to contact with friends and family [83]. However, this study found that there is no moderating effect from communicative social media use. The communicative social media use emphasizes communication with family, friends or acquaintances as a way of obtaining information about the pandemic, which may reduce social isolation caused by loneliness and distance, thereby reducing academic anxiety. As Al-Shuhri’s research found, social media use behavior had a positive impact on the social relationships of participants and their friends and family, allowing them to communicate without restrictions and express their views with ease [84]. Therefore, the differences between communicative and non-communicative social media use in the acquisition of epidemic information when obtaining information on the COVID-19 pandemic need to be further explored.

Otherwise, the use of non-communicative social media use positively regulates the mediating effect of academic anxiety. That is to say, the greater the frequency of non-communicative social media use behavior of undergraduates in the context of the COVID-19 pandemic, the stronger the effect of social isolation on smartphone addiction through academic anxiety. Just as Ostic found that sustained social media use brings about an increase in problematic mobile phone use behavior [85], excessive social media use, entertainment, and gaming are more likely to cause smartphone addiction [86,87]. In the process of undergraduates obtaining news of the COVID-19 pandemic, non-communicative activities have a negative impact on mental health [65], and COVID-19-related news consumption is associated with greater psychological distress [62]. The more exposures associated with the pandemic, the more psychological and behavioral problems the adolescents had [88]. This conclusion also confirms the theoretical view of the stress process model. According to this conclusion, effective psychological intervention strategies should be adopted to reduce the usage time of non-communicative social media among undergraduates. The undergraduates should strengthen social contact with relatives and friends, and then reduce the psychological distress caused by the news exposure of the COVID-19 pandemic.

## 5. Conclusions

This study concludes that social isolation can positively predict smartphone addiction. Academic anxiety plays a mediating role in social isolation influencing smartphone addiction. The more frequent non-communicative social media use behaviors when obtaining news about COVID-19, the greater the impact of social isolation on academic anxiety. Further, the mediating effect of academic anxiety between social isolation and smartphone addiction is affected by non-communicative social media use, namely, there is a moderating mediating role.

In terms of theoretical significance, this study further enriched the theoretical content of smartphone addiction research, and also increased the research literature on smartphone addiction among college students. More importantly, it explored the mechanism of interaction between social isolation, social media use, academic anxiety and smartphone addiction among college students under the background of COVID-19, providing empirical research conclusions for future researchers.

As for practical significance, with the expansion of smartphone functions, the problem of smartphone addiction among undergraduates has become increasingly serious. Based on the background of the COVID-19 outbreak, the quantitative research conclusions also provide a basis for action among college education and teaching administrators and undergraduates themselves. For example, psychological education intervention for college students can alleviate academic anxiety to a certain extent, and then reduce smartphone addiction. Undergraduates themselves can adopt more communicative social media use types to alleviate the negative effects of social isolation in special periods.

Regarding research limitations and future directions, in the first place, the cross-sectional data are used to test the mechanism of social isolation on smartphone addiction, and the true causal relationship cannot be verified, and the impact of COVID-19 is unverifiable. Future research can consider obtaining more practical research conclusions through longitudinal study design. Since the research object is only from one school, the sampling scope needs to be further expanded, and future studies can consider expanding the sampling range and adding students from other academic levels as research samples, such as primary and secondary school students. Secondly, the analysis of the moderating effect of other factor variables could be added. Finally, future research could further explore the factors that affect college students’ smartphone addiction.

## Figures and Tables

**Figure 1 ijerph-19-15903-f001:**
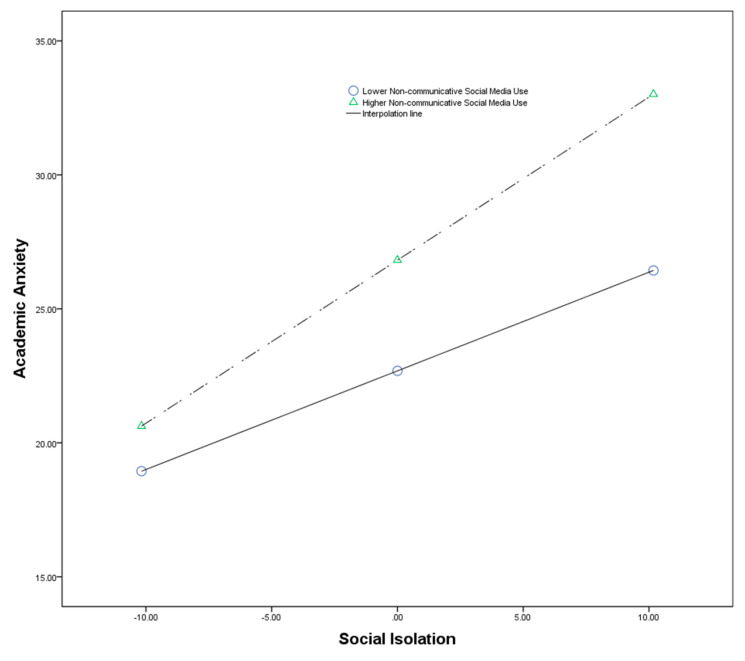
The moderating effect of the relationship between social isolation and academic anxiety.

**Table 1 ijerph-19-15903-t001:** The mean, standard deviation, and correlation coefficient for each variable.

Variable	M	SD	1	2	3	4	5
1. Smartphone addiction	32.14	11.96	1				
2. Communicative social media use	9.12	4.35	0.394 **	1			
3. Non-communicative social media use	9.69	4.18	0.308 **	0.709 **	1		
4. Academic anxiety	24.97	10.84	0.638 **	0.257 **	0.237 **	1	
5. Social isolation	20.25	10.18	0.454 **	0.176 **	0.156 **	0.502 **	1

**. Correlation is significant at the 0.01 level (2-tailed).

**Table 2 ijerph-19-15903-t002:** Harman one-way test statistics for common method biases.

Component	Extration Sums of Squared Loadings			Rotation Sums of Squared of Loadings		
	Eigenvalue	% of Variance	Cumulative %	Eigenvalue	% of Variance	Cumulative %
1	12.18	36.91	36.91	5.66	17.16	17.16
2	3.83	11.60	48.50	5.03	15.24	32.40
3	2.87	8.68	57.19	4.59	13.90	46.30
4	2.13	6.45	63.64	4.08	12.36	58.66
5	1.52	4.60	68.24	2.70	8.19	66.84
6	1.14	3.46	71.70	1.60	4.85	71.70

**Table 3 ijerph-19-15903-t003:** Analysis of the mediating role of academic anxiety.

Variable	Academic Anxiety (M1)			Smartphone Addiction (M2)		
	b	SE	t	b	SE	t
Gender	1.57	1.31	1.20	1.79	1.24	1.44
Age	0.30	0.47	0.64	−1.43	0.45	−3.18
Grade	−1.09	0.73	−1.48	2.22	0.70	3.19
Major	1.61	1.06	1.51	−0.38	1.01	−0.38
Social Isolation	0.55 **	0.05	10.80	0.21 **	0.06	3.80
Academic Anxiety				0.61 **	0.05	11.74
R^2^	0.27			0.46		
F	23.94 **			46.94 **		

** Correlation is significant at the 0.01 level (2-tailed).

**Table 4 ijerph-19-15903-t004:** Analysis of moderating mediating effects (moderating variables = non-communicative social media use).

Variable	Academic Anxiety (M3)			Smartphone Addiction (M4)		
	b	SE	t	b	SE	t
Gender	0.95	1.28	0.74	1.00	1.20	0.83
Age	0.21	0.46	0.46	−1.50	0.43	−3.49
Grade	−1.34	0.71	−1.88	1.77	0.67	2.64
Major	1.85	1.03	1.79	−0.04	0.97	−0.04
Social Isolation	0.49 **	0.05	9.67	0.19 **	0.05	3.56
Academic Anxiety				0.54 **	0.05	10.57
Non-communicative social media use	0.47 **	0.12	4.04	0.64 **	0.11	5.67
Social Isolation × Non-communicative social media use	0.03 **	0.01	2.71	0.01	0.01	0.93
R^2^	0.31			0.51		
F	21.42 **			42.51 **		

**. Correlation is significant at the 0.01 level (2-tailed).

**Table 5 ijerph-19-15903-t005:** Conditional indirect effects analysis.

Moderating Variables	Indirect Effect	BootSE	BootLLCI	BootULCI
Low Non-communicative social media use	0.20	0.05	0.10	0.31
Middle Non-communicative social media use	0.27	0.04	0.19	0.35
High Non-communicative social media use	0.33	0.05	0.24	0.44

**Table 6 ijerph-19-15903-t006:** Analysis of moderating mediating (moderating variable = communicative social media use).

Variable	Academic Anxiety (M5)			Smartphone Addiction (M6)		
	b	SE	t	b	SE	t
Gender	1.28	1.29	0.99	1.62	1.23	1.32
Age	0.32	0.46	0.70	−1.40	0.44	−3.18
Grade	−1.42	0.73	−1.95	1.91	0.70	2.74
Major	1.73	1.05	1.65	−0.19	1.00	−0.19
Social Isolation	0.51 **	0.05	10.18	0.20 **	0.06	3.66
Academic Anxiety				0.57 **	0.05	10.98
Communicative social media use	0.46 **	0.12	3.72	0.42 **	0.12	3.50
Social Isolation × Communicative social media use	0.003	0.01	0.25	0.01	0.01	0.68
R^2^	0.54			0.48		
F	19.69 **			37.92 **		

**. Correlation is significant at the 0.01 level (2-tailed).

## Data Availability

The data presented in this study are available on request from the corresponding author.

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
