# Peer review of "The Impact of Undergraduates’ Social Isolation on Smartphone Addiction: The Roles of Academic Anxiety and Social Media Use"

_ijerph, 2022, doi:10.3390/ijerph192315903_

Round 1

Reviewer 1 Report

This manuscript is dealing with a topic that is hardly interesting to deal with at this time. Covid was a strong hit to the world population and the thesis in this article are by now obvious.

A major concern is plagscan (5% is acceptable) but this manuscript has 20.8% plagiarism detected. Please see the attached file. Red is direct copy-paste.

Blue is altered text.

On the other hand, the authors have put a lot of effort into the literature review.. and that is by far the strongest point of this article. Also for pointing out, the number of respondents is preaty good.

Overall, this is an interesting work, but it should end here since the material is heavily recicled.

Author Response

Dear reviewer,

Thank you for your review and evaluation of our manuscript. Regarding the duplication rate of the manuscript, we have adjusted and modified it according to the report you provided.

Best regards

Reviewer 3 Report

The article analyzes the associations between social isolation of university students, the use of social networks to obtain information about the pandemic, academic anxiety and smartphone addiction. I would like to thank the authors for the work done and the Editorial Team of the journal for the opportunity to act as a reviewer.

Some considerations:

• Excessively long title, it is advisable to shorten it.

• A pandemic context would not be necessary to explain significant relationships between mobile phone addiction and social isolation. This relationship would be more clearly reflected after a study before and during the pandemic.

• Social isolation is not a “new” problem that has emerged from the pandemic. It would be advisable to indicate figures prior to the pandemic to show a comparison with the percentages indicated in the article, so that the increase in social isolation could be verified.

• It is advisable to probably define what an addiction is.

• Beware of “definitive” value judgments, the authors point out that the COVID-19 pandemic has destroyed social integration.

• Excess of bibliographical references (almost six pages), only in the justification presenting 105 references. It is advisable to make a selection of the most outstanding investigations for the study. On the other hand, we present a high number of references from the last three years, this update is relevant.

• It would be advisable to explain the reason for the differences in the selection of the sample according to their gender and academic level. The size of the population should also be noted. The type of test used for the selection of participants is not clear.

• The instruments could be included, it is not clear, for example, the number of elements that make up each of them (do they include dimensions to encompass each of the elements?). It would be advisable to recommend more statistical validation indexes of the instruments. Likert-type scales need to be explained in more detail.

• It would be advisable to include a section on procedure that includes, among other things, ethical considerations of the research or statistical package used to obtain the results. Similarly, a section on the analysis of the data to be developed should be included.

• The results of the Tables could be explained further, for example Table 1.

• The Discussion includes the main results achieved and their relationship with those obtained in other investigations.

• Includes limitations and proposals for future research.

• Short conclusion, it is advisable to expand. For example, the limitations of the research could be included here.

Round 2

Reviewer 1 Report

Small tweaks are done to the original manuscript for the overall passing score. Further articles should use this as preliminary research, but be aware of plug scan software and a marginal edge of 5% for acceptance of valid scientific work.

Reviewer 3 Report

The authors have taken into consideration the reviews carried out. Congratulations for the work developed!